# WFM: 3D Wavelet Flow Matching for Ultrafast Multi-Modal MRI Synthesis

**Yalcin Tur**[1]                                                    YALCINTR@STANFORD.EDU

**Mihajlo Stojkovic**[1]                                             MSTOJKOV@STANFORD.EDU

**Ulas Bagci**[2]                                             ULAS.BAGCI@NORTHWESTERN.EDU

[1] *Department of Computer Science, Stanford University, Stanford, CA, USA.*

[2] *Machine and Hybrid Intelligence Lab, Feinberg School of Medicine, Northwestern University, Chicago, IL, USA.*

**Editors:** Accepted for publication at MIDL 2026

## Abstract

Diffusion models have achieved remarkable quality in multi-modal MRI synthesis, but their computational cost (hundreds of sampling steps and separate models per modality) limits clinical deployment. We observe that this inefficiency stems from an unnecessary starting point: diffusion begins from pure noise, discarding the structural information already present in available MRI sequences. We propose WFM (Wavelet Flow Matching), which instead learns a direct flow from an *informed prior*, the mean of conditioning modalities in wavelet space, to the target distribution. Because the source and target share underlying anatomy and differ primarily in contrast, this formulation enables accurate synthesis in just 1-2 integration steps. A single 82M-parameter model with class conditioning synthesizes all four BraTS modalities (T1, T1c, T2, FLAIR), replacing four separate diffusion models totaling 326M parameters. On BraTS 2024, WFM achieves 26.8 dB PSNR and 0.94 SSIM, within 1-2 dB of diffusion baselines, while running 250-1000x faster (0.16-0.64s vs. 160s per volume). This speed-quality trade-off makes real-time MRI synthesis practical for clinical workflows. Code is available at https://github.com/yalcintur/WFM.

**Keywords:** MRI synthesis, flow matching, wavelet transform, diffusion models, multi-modal imaging

## 1. Introduction

Accurate brain tumor analysis requires multiple MRI contrasts (T1-weighted, T1 with contrast enhancement (T1c), T2-weighted, and T2-FLAIR), each revealing distinct tissue properties essential for segmentation and treatment planning (Baid et al., 2021). In practice, however, complete acquisitions are frequently unavailable: scan time constraints, patient motion, or imaging artifacts often leave one or more modalities missing. Synthesizing these missing sequences from available ones has therefore become an active research focus, with recent diffusion-based methods achieving impressive fidelity. Yet a fundamental barrier remains: these methods are too slow for clinical use.

Diffusion models have set the quality standard for medical image synthesis (Friedrich et al., 2024a; Kim and Park, 2024; Özbey et al., 2023). The conditional Wavelet Diffusion Model (cWDM) demonstrated that operating in wavelet space enables memory-efficient 3D synthesis at full resolution (Friedrich et al., 2024a). However, these methods inherit a fundamental inefficiency from the diffusion framework: they begin from pure Gaussian noise.

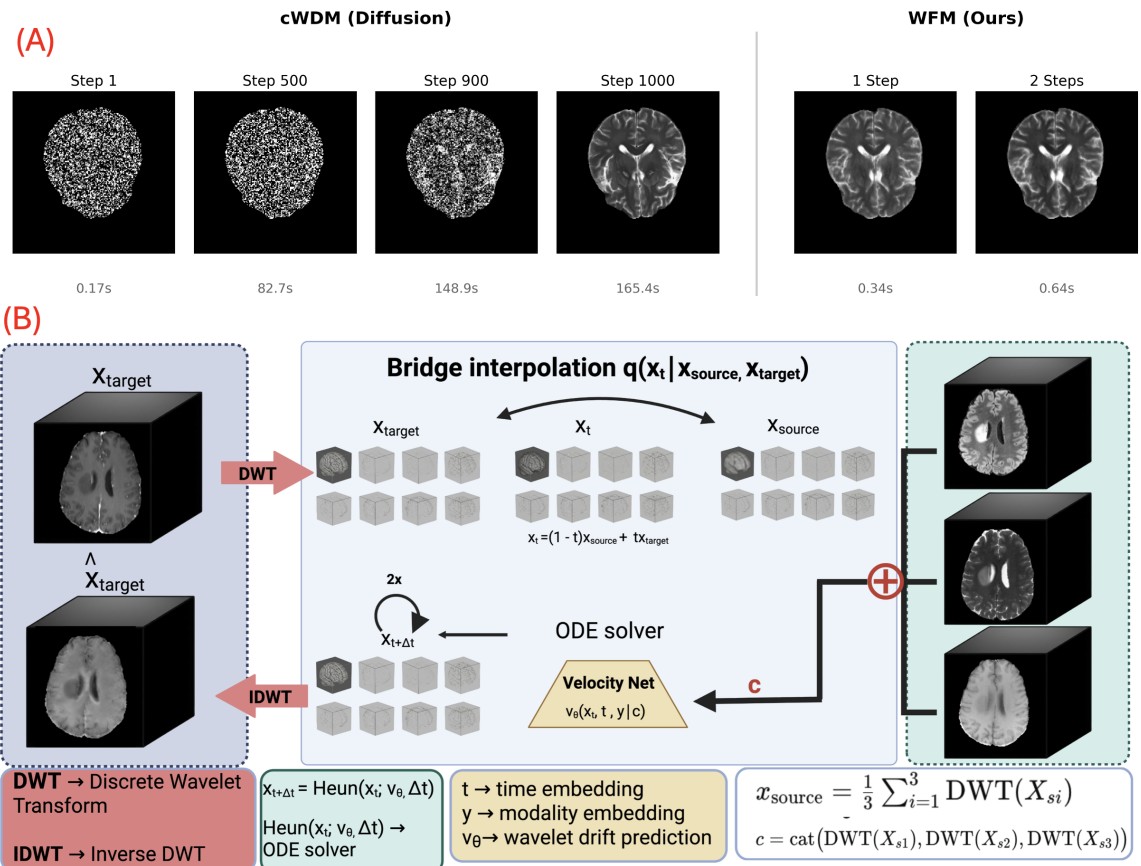

Figure 1: **(A)** Comparison of synthesis speed. cWDM requires 1000 denoising steps (165.4s) to transform noise into a valid MRI. WFM produces comparable quality in 1-2 steps (0.34-0.64s) by starting from an informed prior rather than noise. **(B)** Overview of WFM. Three conditioning modalities are transformed to wavelet space and averaged to form the informed source $\mathbf{x}_{\text{source}}$; their concatenation forms the condition $\mathbf{c}$. The velocity network predicts the drift from source to target, conditioned on timestep $t$ and target modality $y$. An ODE solver (Euler or Heun) integrates the velocity field in 1-2 steps. The inverse wavelet transform (IDWT) produces the final synthesized volume $\widehat{\mathbf{X}}_{\text{target}}$.

Reconstructing meaningful anatomical structure from noise requires iterative refinement: typically 1000 denoising steps, translating to approximately 160 seconds per volume. Moreover, because each target modality defines a distinct generation task, existing approaches train separate models for each contrast, resulting in four independent networks totaling 326M parameters for the BraTS protocol. Together, these costs place diffusion-based synthesis outside the realm of real-time clinical deployment.

Our key observation is that the diffusion starting point is unnecessarily uninformative. In multi-modal MRI synthesis, source and target modalities capture the same anatomy with different contrast weightings: T1 and T2 images of the same brain differ in intensity

mapping, not in structure. The mean of available modalities, therefore, provides a strong prior: it already contains ventricle boundaries, cortical folding, and lesion locations. Rather than reconstructing this structure from noise, we can learn a direct transformation from the **informed prior** to the target contrast. This insight aligns with recent advances in flow matching (Lipman et al., 2023) and bridge-based diffusion (Liu et al., 2023; Li et al., 2023), which demonstrate that informative starting points enable few-step generation.

We propose **WFM (Wavelet Flow Matching)**, a method that combines these principles with 3D wavelet-domain processing for efficient multi-modal MRI synthesis. Our contributions are threefold:

1. **Informed-prior flow matching.** We formulate synthesis as learning a velocity field from the mean of conditioning modalities (in wavelet space) to the target distribution. Because the prior already encodes anatomical structure, the model learns only the contrast transformation, enabling accurate synthesis in 1-2 integration steps rather than 1000.

2. **Unified multi-modality architecture.** A single 82M-parameter network with class conditioning synthesizes all four BraTS modalities, replacing four separate models (326M total) while sharing learned anatomical representations across tasks.

3. **Clinically viable speed.** WFM synthesizes a full 240×240×155 volume in 0.16-0.64 seconds, a 250-1000x speedup over cWDM (**Figure** 1A), bringing MRI synthesis into the realm of interactive clinical workflows.

## 2. Related Work

### 2.1. Diffusion Models for Medical Image Synthesis

Diffusion models have become the dominant paradigm for high-fidelity medical image synthesis, displacing GANs where quality is paramount. Friedrich et al. (2024b) addressed the memory challenge of 3D synthesis by operating in wavelet space, and their conditional extension cWDM (Friedrich et al., 2024a) demonstrated state-of-the-art quality on BraTS. Kim and Park (2024) combined SPADE normalization with latent diffusion; Özbey et al. (2023) proposed SynDiff for unsupervised translation. A separate line of work explores text-guided MRI generation (Wang et al., 2024), which offers flexible conditioning but addresses a different task than our image-to-image translation. Despite architectural innovations, all these methods require hundreds to thousands of sampling steps. Acceleration techniques (DDIM (Song et al., 2020), progressive distillation (Salimans and Ho, 2022), consistency models (Song et al., 2023)) reduce step counts but retain noise as the starting point. WFM instead changes the starting point itself: by beginning from an informed prior, single-step integration becomes accurate without distillation.

### 2.2. Flow Matching and Bridge Methods

Flow matching (Lipman et al., 2023) learns continuous-time flows between distributions, offering simpler training objectives than diffusion. Several works exploit informative starting distributions: I²SB (Image-to-Image Schrödinger Bridge) Liu et al. (2023) formulates

translation as a Schrödinger bridge, BBDM Li et al. (2023) uses Brownian bridge processes, and DSBM Shi et al. (2023) provides theoretical foundations. WFM makes a simplifying assumption suited to multi-modal MRI: because source and target share anatomy and differ only in contrast, a linear interpolation path suffices, yielding single-step Euler integration without complex bridge formulations.

### 2.3. GAN-based Medical Image Translation

Before diffusion, GANs dominated medical synthesis (Pix2Pix (Isola et al., 2017), Cycle-GAN (Zhu et al., 2017), ResViT Dalmaz et al. (2022)), offering fast inference but suffering from training instability in 3D (Saad et al., 2024). WFM achieves GAN-like speed through flow matching while avoiding adversarial training. For memory efficiency, we inherit the wavelet-domain formulation of (Friedrich et al., 2024b): the 3D Haar transform reduces spatial dimensions while preserving information losslessly, enabling full-resolution synthesis. Finally, we adopt a unified multi-task architecture with class conditioning, reducing parameters from 326M (four cWDM models) to 82M while sharing anatomical representations (Chartsias et al., 2018).

## 3. Methods

**Problem Formulation**. We consider the task of synthesizing a missing MRI modality from a set of available sequences. Let $\{\mathbf{X}_{s_1}, \mathbf{X}_{s_2}, \mathbf{X}_{s_3}\} \in \mathbb{R}^{D \times H \times W}$ denote the three out of four BraTS modalities (T1, T1c, T2, FLAIR), where each $\mathbf{X}_{s_i} \in \mathbb{R}^{D \times H \times W}$ is a 3D volume with depth $D$, height $H$, and width $W$. Given three available modalities $\{\mathbf{X}_{s_1}, \mathbf{X}_{s_2}, \mathbf{X}_{s_3}\}$ as conditioning inputs, our goal is to synthesize the missing target $X_t$. Prior diffusion-based approaches train separate models for each target modality, yielding four independent networks. We instead learn a single unified model $f_\theta$ that synthesizes any of the four modalities, using a class label $y \in \{0, 1, 2, 3\}$ to specify which modality to generate. However, this formulation assumes exactly three conditioning modalities are available, a constraint we discuss in the Appendix. The key departure from diffusion-based synthesis is the choice of starting distribution. Rather than generating $X_t$ by denoising from Gaussian noise, we construct an informed prior from the conditioning modalities and learn a direct transformation to the target. The following sections formalize this approach.

### 3.1. Flow Matching with Informed Prior

**Background.** Standard diffusion models define a forward process that progressively corrupts data toward isotropic Gaussian noise, then learn to reverse this process. Generation proceeds by sampling from $\mathcal{N}(0, \mathbf{I})$ and iteratively denoising. The limitation is fundamental: pure noise contains no information about the target, so the model must reconstruct all structure from scratch, a task requiring many refinement steps. Flow matching (Lipman et al., 2023) offers an alternative formulation. Rather than learning a denoising process, flow matching learns a velocity field $\mathbf{v}(x, t)$ that transports samples from a source distribution to a target distribution along continuous paths. The framework is flexible: any differentiable path connecting the source and target defines a valid flow.

**Informed source construction.** Our key insight is that the source distribution need not be uninformative noise. In multi-modal MRI synthesis, the conditioning modalities $\{\mathbf{X}_{s_1}, \mathbf{X}_{s_2}, \mathbf{X}_{s_3}\}$ already encode the patient's anatomy: the same ventricles, cortical folds, and lesions that will appear in the target $X_t$, differing only in intensity mapping. We construct an informed source by averaging the conditioning modalities in wavelet space:

$$\mathbf{x}_{\text{source}} = \frac{1}{3} \sum_{i=1}^{3} \text{DWT}(\mathbf{X}_{s_i}), \tag{1}$$

where DWT denotes the 3D discrete wavelet transform (detailed in Appendix B). The mean operation is motivated by two considerations. First, averaging suppresses modality-specific intensity variations while preserving shared anatomical structure: edges, boundaries, and spatial organization that are consistent across sequences. Second, the mean provides a simple, parameter-free aggregation that introduces no additional learned components. Alternative aggregation strategies (learned attention, concatenation with reduction, or modality-specific weighting) could potentially improve performance but would increase model complexity. We find that simple averaging suffices for the BraTS synthesis task (see **Figure 1** for the overall pipeline).

**Linear interpolation path.** Given source $\mathbf{x}_{\text{source}}$ and target $\mathbf{x}_{\text{target}} = DWT(\mathbf{X_t})$, we define a linear interpolation: $\mathbf{x}_t = (1 - t)\mathbf{x}_{\text{source}} + t\mathbf{x}_{\text{target}}$, where $t \in [0, 1]$. This path has a constant velocity:

$$\mathbf{v} = \frac{d\mathbf{x}_t}{dt} = \mathbf{x}_{\text{target}} - \mathbf{x}_{\text{source}}. \tag{2}$$

The linear path is the simplest choice and corresponds to optimal transport with quadratic cost when source and target are point masses. For MRI synthesis, where source and target are structurally aligned, this direct path is appropriate: there is no need for the curved trajectories that optimal transport would prescribe for distant, misaligned distributions. In practice, the learned velocity $\mathbf{v}$ is an approximation, but the structural alignment between source and target ensures that errors are small, enabling accurate synthesis in 1-2 steps rather than hundreds.

### 3.2. Training Objective

We train a neural network $f_\theta$ to predict the constant velocity $\mathbf{v} = \mathbf{x}_{\text{target}} - \mathbf{x}_{\text{source}}$. The training objective is a simple regression loss:

$$\mathcal{L} = \mathbb{E}_{t,\boldsymbol{\epsilon}} \left[ \| f_\theta(\tilde{\mathbf{x}}_t, \mathbf{c}, t, y) - (\mathbf{x}_{\text{target}} - \mathbf{x}_{\text{source}}) \|^2 \right] \tag{3}$$

where $t \sim \mathcal{U}(0, 1)$ is a uniformly sampled timestep, $y \in \{0, 1, 2, 3\}$ is the target modality class, and $\mathbf{c}$ is the conditioning information (defined below).

**Conditioning representation.** The condition $\mathbf{c}$ consists of the concatenated wavelet coefficients of all three source modalities. This provides the model with full access to each conditioning modality separately, enabling it to learn modality-specific contrast relationships rather than relying solely on the averaged source.

**Stochastic regularization.** During training, we add noise to the interpolant:

$$\tilde{\mathbf{x}}_t = \mathbf{x}_t + \sigma \sqrt{t(1-t)}\boldsymbol{\epsilon}, \quad \boldsymbol{\epsilon} \sim \mathcal{N}(0, \mathbf{I}). \tag{4}$$

The noise schedule $\sigma$ has a specific motivation. At the endpoints ($t = 0$ and $t = 1$), the noise vanishes, ensuring that the model sees clean source and target samples. At intermediate timesteps, noise is maximal at $t = 0.5$, where the interpolant is equidistant from both endpoints and perturbations are least disruptive to the learning signal. This regularization serves two purposes. First, it prevents the model from memorizing the deterministic interpolation path, encouraging generalization to unseen source-target pairs. Second, it provides robustness during inference: small errors in the predicted velocity do not compound catastrophically because the model has been trained on perturbed trajectories. We set $\sigma = 0.5$ based on preliminary experiments. Higher values degrade quality by obscuring the target velocity signal; lower values reduce regularization benefit.

### 3.3. Unified Multi-Modality Architecture

A naive approach to multi-modal synthesis trains separate models for each target modality: four networks for the BraTS protocol, totaling 326M parameters (81.5M × 4). This is inefficient: the networks learn redundant anatomical representations, and knowledge cannot transfer across synthesis tasks. We instead train a single unified model $f_\theta$ (3D U-Net with class conditioning) that synthesizes any target modality, specified by a class label $y$. This reduces total parameters to 81.5M (a 4x reduction) while enabling the network to share learned representations across all synthesis directions. The class label $y$ is embedded and added to the timestep embedding: $\mathbf{e} = \text{TimeEmbed}(t) + \text{ClassEmbed}(y)$. This vector modulates the network through adaptive normalization: each residual block scales and shifts its normalized activations based on $\mathbf{e}$, enabling the same weights to produce different behaviors for different target modalities and timesteps. The model takes as input the concatenation of the noisy interpolant $\tilde{\mathbf{x}}_t$ (8 channels) with all three conditioning modalities in wavelet space (24 channels), totaling 32 input channels. 3D U-Net with base channels 64, channel multipliers (1, 2, 2, 4, 4), 2 residual blocks per level. Total parameters: 81.5M.

At inference time, we synthesize a target modality by integrating the learned velocity field from source to target. The number of function evaluations (NFE), i.e. forward passes through the network, determines both quality and speed. We consider two ODE solvers: Euler (first-order) and Heun (second-order Runge-Kutta). For single-step Euler (NFE=1), this directly yields the target $\hat{\mathbf{x}}_{\text{target}} = \mathbf{x}_{\text{source}} + f_\theta(\mathbf{x}_{\text{source}}, \mathbf{c}, t = 0, y)$. For higher accuracy, Heun's method uses a predictor-corrector scheme (2 forward calls per step):

$$\mathbf{v}_1 = f_\theta(\mathbf{x}_t, \mathbf{c}, t, y), \tag{5}$$

$$\tilde{\mathbf{x}}_{t+\Delta t} = \mathbf{x}_t + \mathbf{v}_1 \cdot \Delta t, \tag{6}$$

$$\mathbf{v}_2 = f_\theta(\tilde{\mathbf{x}}_{t+\Delta t}, \mathbf{c}, t + \Delta t, y), \tag{7}$$

$$\mathbf{x}_{t+\Delta t} = \mathbf{x}_t + \frac{\mathbf{v}_1 + \mathbf{v}_2}{2} \cdot \Delta t. \tag{8}$$

The final output is obtained by applying the inverse wavelet transform.

## 4. Experiments

**Dataset and preprocessing:** We evaluate on BraTS 2024 (Baid et al., 2021), a multi-institutional dataset of brain MRI volumes from glioma patients. Each case includes four co-registered modalities: T1-weighted (T1), T1 with gadolinium contrast enhancement (T1c),

T2-weighted (T2), and T2-FLAIR. Volumes are resampled to $1\text{mm}^3$ isotropic resolution with dimensions $240 \times 240 \times 155$ voxels. We use the official BraTS 2024 split: 1,032 volumes for training and 219 for validation. The validation set serves as our test set since BraTS withholds ground truth for the official test partition. All reported metrics are computed on this 219-volume validation set. Each modality is independently normalized to zero mean and unit variance within the brain mask. We do not apply additional intensity standardization (e.g., histogram matching) to preserve the natural contrast variations that the synthesis model must learn to handle.

**Implementation details and baselines:** Training uses AdamW optimizer with learning rate $10^{-5}$, batch size 4, for 50K iterations on a single NVIDIA A100. During training, the target modality is randomly selected each step. We use $\sigma = 0.5$ for regularization noise. We compare against three baselines: (i) cWDM (Friedrich et al., 2024a), a wavelet-space conditional diffusion model that requires 1000 sampling steps and uses four separate models (one per target modality); (ii) a recent flow-matching baseline (CFM) adapted from Chang et al. (Chang et al., 2025), configured to accept three conditioning modalities for a fair comparison; and (iii) a 3D Pix2Pix baseline adapted from (Isola et al., 2017), extended with class-conditioning to generate all four target modalities using a single generator. All methods use the same preprocessing and the same BraTS split, and we report PSNR/SSIM on the validation set within the brain mask.

**Metrics and their potential limitations:** We report two complementary metrics computed within the brain mask: PSNR and SSIM. PSNR measures pixel-wise reconstruction accuracy in decibels. Higher is better. SSIM measures perceptual similarity based on luminance, contrast, and structure. Range [0, 1]; higher is better. Both metrics are computed per-volume and averaged across the validation set. PSNR and SSIM assess pixel-level fidelity but do not directly measure clinical utility. A more complete evaluation would include: (1) perceptual metrics like LPIPS or FID adapted for 3D medical images, (2) downstream task performance (e.g., tumor segmentation accuracy using synthetic vs. real modalities), and (3) radiologist evaluation of diagnostic quality. We discuss these limitations in the Appendix.

### 4.1. Main Results

Table 1 presents the quantitative comparison between WFM and cWDM across all four BraTS modalities. Overall, WFM achieves 26.8 dB average PSNR and 0.94 SSIM with 1-2 function evaluations, compared to cWDM's 28.4 dB PSNR and 0.95 SSIM with 1000 evaluations. The quality gap of 1.6 dB PSNR represents a meaningful but bounded degradation. As shown in Figure 2 (also corresponding sagittal views in Appendix see Figure 3), this translates to outputs that preserve major anatomical structures while showing reduced detail primarily in challenging regions such as tumor cores with unpredictable enhancement patterns. Performance varies across modalities (T1c gap: 0.78 dB; T1 gap: 2.12 dB), reflecting differing prediction difficulty. Heun's method (NFE=4) provides only marginal improvement over single-step Euler (26.80 vs. 26.72 dB), confirming that the learned velocity field is approximately constant. See Appendix C.4 for detailed analysis.

**Unified vs. separate models.** The single unified WFM model matches the quality achievable with four hypothetically separate WFM models (we verified this in preliminary

Table 1: Comparison on BraTS 2024 validation set. WFM achieves competitive quality with fewer parameters and significantly faster inference. *CFM is our reimplementation of* Chang et al. (2025).

| Method | Models | Params | NFE | T1 | T1c | T2 | FLAIR | Time |
|--------|--------|--------|-----|-----|-----|-----|-------|------|
| *PSNR (dB) ↑* | | | | | | | | |
| cWDM | 4 | 326M | 1000 | 29.74 | 27.31 | 28.86 | 27.83 | 160s |
| CFM | 1 | 94M | 1 | 25.34 | 26.78 | 25.96 | 25.35 | 0.16s |
| Pix2Pix3D | 1 | 55.9M | 1 | 25.25 | 24.52 | 24.22 | 23.80 | 0.098s |
| **WFM** (Euler, 1) | 1 | 82M | 1 | 27.41 | 26.40 | 27.15 | 25.91 | 0.16s |
| **WFM** (Heun, 1) | 1 | 82M | 1 | 27.51 | 26.39 | 27.24 | 25.91 | 0.32s |
| **WFM** (Heun, 2) | 1 | 82M | 2 | 27.62 | 26.53 | 26.90 | 26.13 | 0.64s |
| *SSIM ↑* | | | | | | | | |
| cWDM | 4 | 326M | 1000 | 0.956 | 0.936 | 0.952 | 0.935 | 160s |
| CFM | 1 | 94M | 1 | 0.924 | 0.912 | 0.924 | 0.922 | 0.16s |
| Pix2Pix3D | 1 | 55.9M | 1 | 0.899 | 0.882 | 0.879 | 0.855 | 0.098s |
| **WFM** (Euler, 1) | 1 | 82M | 1 | 0.947 | 0.930 | 0.943 | 0.928 | 0.16s |
| **WFM** (Heun, 1) | 1 | 82M | 1 | 0.948 | 0.930 | 0.944 | 0.928 | 0.32s |
| **WFM** (Heun, 2) | 1 | 82M | 2 | 0.948 | 0.932 | 0.944 | 0.930 | 0.64s |

experiments, finding $\leq 0.1$ dB difference). This validates that class conditioning effectively specializes the shared architecture for each synthesis direction, with the added benefit of cross-task representation sharing.

**Statistical considerations.** Validation-set standard deviations are substantial ($\pm 2$–$3$ dB) due to patient variability; the WFM-cWDM gap is consistent and significant.

### 4.2. Efficiency Analysis

Table 2 summarizes the computational efficiency of WFM compared to cWDM. WFM achieves 250-1000x speedup depending on the integration scheme. The speedup is multiplicative: each cWDM denoising step requires one forward pass through an 81.5M parameter network, and 1000 steps accumulate to 160 seconds per volume. WFM's informed prior reduces this to 1-4 forward passes. WFM's unified model replaces four separate cWDM models (326M total), a $4\times$ reduction impacting storage and deployment complexity. Per-forward-pass memory is similar ( 12GB on A100); the primary benefit is wall-clock time.

### 4.3. Qualitative Results

Figure 2 (and 4 in the Appendix D) present representative synthesis results across multiple test cases. We analyze both successful syntheses and failure modes.

**Anatomical fidelity and successful tumor synthesis.** WFM consistently preserves major anatomical structures across all modalities. Ventricular boundaries, cortical folding

Table 2: Efficiency comparison. WFM achieves 4× parameter reduction and 250-1000x speedup compared to cWDM. Each forward pass takes ∼160ms on an NVIDIA A100.

| Method | Models | Params | Fwd Calls | Time | Speedup |
|---|---|---|---|---|---|
| cWDM (Friedrich et al., 2024a) | 4 separate | 326M | 1000 | 160s | 1× |
| **WFM** (Euler, 1) | 1 unified | 82M | 1 | 0.16s | 1000× |
| **WFM** (Heun, 1) | 1 unified | 82M | 2 | 0.32s | 500× |
| **WFM** (Heun, 2) | 1 unified | 82M | 4 | 0.64s | 250× |

patterns, and white-gray matter interfaces align closely with ground truth. This confirms that the informed prior successfully transfers structural information, and the learned velocity field accurately transforms contrast without distorting anatomy. Figure 2(right) (Samples 46 and 72) demonstrates that WFM can faithfully synthesize challenging pathological regions. Tumor boundaries, peritumoral edema, and heterogeneous enhancement patterns are preserved when sufficient structural cues exist in the conditioning modalities. The T2/FLAIR hyperintensity that delineates edema transfers effectively to synthesized T1c, enabling plausible enhancement prediction.

**Failure modes.** Figure 2 (Sample 30) illustrates under-detailed tumor cores where T1c enhancement depends on blood-brain barrier disruption, which is not observable in non-contrast sequences.

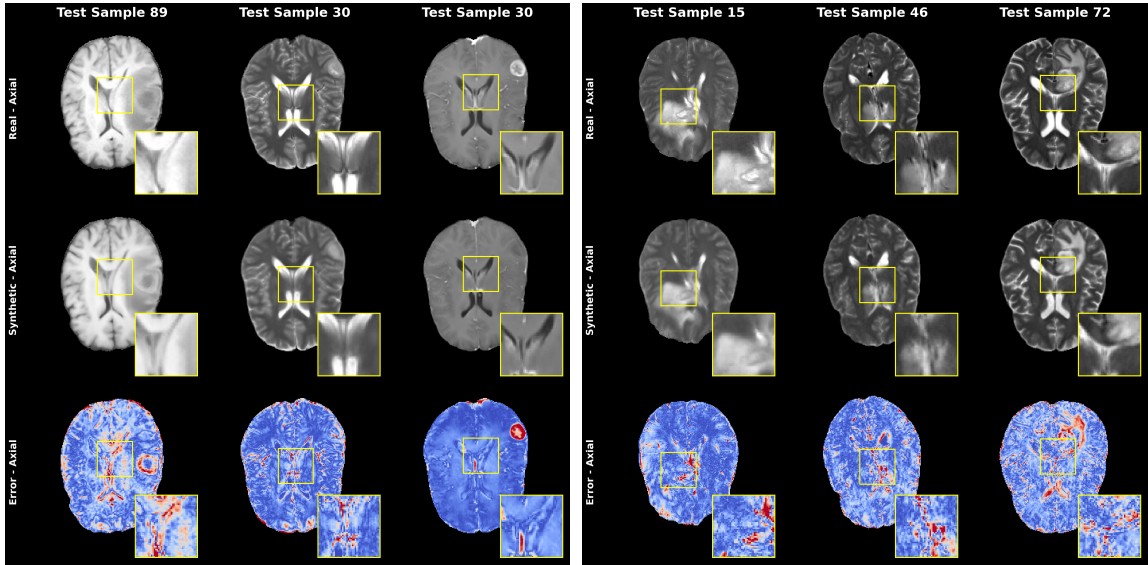

Figure 2: Qualitative results on axial slices with zoomed regions and error maps. **Left:** Sample 89 shows successful synthesis; Sample 30 illustrates a failure case with less detailed tumor regions. **Right:** Samples 15, 46, and 72 demonstrate accurate preservation of tumor boundaries and enhancement patterns. Error maps use a blue-to-red colormap (red = higher error).

## 5. Discussion and Concluding Remarks

We presented WFM, a method that reframes multi-modal MRI synthesis as flow matching from an informed prior rather than denoising from noise. Because source and target modalities share underlying anatomy, the mean of conditioning modalities provides a starting point requiring only contrast transformation. The result is a 250-1000x speedup (0.16-0.64s vs. 160s per volume) with a single 82M-parameter model replacing four separate diffusion networks, at a cost of 1.6 dB PSNR (26.8 vs. 28.4 dB). More broadly, WFM illustrates that task-specific structure can dramatically reduce computational requirements: the assumption that generation must begin from noise is a modeling choice, not a necessity.

Future work includes extending to cross-modality pairs with weaker structural alignment (CT-to-MRI), downstream validation on tumor segmentation, and uncertainty quantification for safe clinical deployment. We also note that fast MR harmonization methods (e.g., HACA3 (Zuo et al., 2023)) may serve as strong efficiency-oriented translation baselines, though a fair comparison would require non-trivial adaptation.

**Why Informed Priors Enable Few-Step Synthesis.** The speedup (from 1000 steps to 1-2) stems from initialization: conditioning modalities already encode anatomy, so the mean provides a starting point structurally close to the target. The model learns contrast transformation rather than structural reconstruction, explaining why NFE>2 provides diminishing returns (Table 3).

**Quality-speed trade-off.** WFM achieves 26.8 dB PSNR versus cWDM's 28.4 dB, a 1.6 dB gap. Whether this is acceptable depends on application: for real-time clinical review, sub-second synthesis with 0.94 SSIM may be preferable to 160-second waits; for maximum-fidelity research, the gap matters more. The gap varies by modality (T1c: 0.78 dB; T1: 2.12 dB), reflecting varying difficulty of contrast prediction.

**Clinical relevance of sub-second synthesis.** While WFM trades approximately 1.6 dB PSNR for a 250–1000× speedup, this trade-off is application-dependent. In clinical imaging, latency is not merely a convenience but a first-order quality attribute. In time-critical settings such as neurosurgical planning, emergency triage, and intensive care, delays on the order of minutes are prohibitive, whereas sub-second synthesis enables on-demand visualization of missing contrasts. Similarly, interactive radiologist review benefits from immediate synthesis when a modality is unavailable or corrupted, and resource-constrained settings benefit from single-pass inference without dedicated compute infrastructure. In these scenarios, near–real-time synthesis can enable workflows that iterative diffusion methods cannot support, despite their marginally higher fidelity.

**Limitations.** Our evaluation has several limitations: (1) Single dataset (BraTS 2024 glioma patients); generalization to other pathologies is unvalidated. (2) No downstream evaluation: we do not test whether synthetic modalities improve segmentation. (3) Fixed conditioning: WFM assumes exactly three available modalities. (4) Failure modes: WFM struggles with unpredictable enhancement patterns in tumor cores, where T1c contrast depends on blood-brain barrier disruption not observable in non-contrast sequences. Extended discussion appears in the Appendix.

## Acknowledgments

This study is partially supported by the following NIH grants: R01HL171376 and U01CA268808.

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

Table 3: Effect of number of function evaluations (NFE). NFE=2 with Heun's method achieves optimal quality.

| Solver | NFE | T1 | T1c | T2 | FLAIR | Avg PSNR |
|--------|-----|-------|-------|-------|-------|----------|
| Euler  | 1   | 27.41 | 26.40 | 27.15 | 25.91 | 26.72 |
| Heun   | 1   | 27.51 | 26.39 | 27.24 | 25.91 | 26.76 |
| Heun   | 2   | 27.62 | 26.53 | 26.90 | 26.13 | 26.80 |

## Appendix A. Ablation Studies

We conduct ablation experiments to validate key design choices. Unless otherwise specified, ablations use Heun integration with 1 step (NFE=2). Table 3 shows that NFE=1-2 achieves optimal quality. Additional steps provide no benefit, consistent with our formulation where the model learns an approximately constant velocity field.

## Appendix B. Wavelet-Domain Processing

Operating directly on full-resolution 3D volumes (240×240×155 voxels at 32-bit precision $\approx$ 34 MB per volume) is memory-prohibitive for deep networks with batch processing and intermediate activations. Following (Friedrich et al., 2024b,a; Phung et al., 2023), we process volumes in wavelet space, where spatial dimensions are reduced while information is preserved.

**3D Discrete Wavelet Transform.** The 3D Haar wavelet transform decomposes a volume $\mathbf{X} \in \mathbb{R}^{D \times H \times W}$ into 8 subbands:

$$\mathrm{DWT}(\mathbf{X}) \in \mathbb{R}^{8 \times \frac{D}{2} \times \frac{H}{2} \times \frac{W}{2}} \tag{9}$$

reducing spatial dimensions by 2× in each axis while preserving multi-scale information through the LLL (low-frequency) and high-frequency subbands. Each subband captures different frequency content: LLL: Low-frequency approximation (coarse structure, smooth regions), LLH, LHL, HLL: Mixed frequency (edges along one axis), LHH, HLH, HHL: Higher frequency (edges along two axes), HHH: Highest frequency (corners, fine texture). The transform is orthogonal and perfectly invertible: $IDWT(DWT(\mathbf{X})) = \mathbf{X}$ with no information loss. This distinguishes wavelet processing from learned latent spaces, which introduce reconstruction error.

**Choice of Haar and Processing pipeline.** We use Haar wavelets for their computational simplicity: the transform requires only additions and subtractions, with no floating-point multiplications. More sophisticated wavelet families (Daubechies, biorthogonal) provide better frequency localization but increase computational cost without clear benefit for our synthesis task, where the learned network can compensate for transform limitations. All operations (source construction, interpolation, velocity prediction, and ODE integration) occur in wavelet space. Only the final output is transformed back to image space via IDWT for evaluation and visualization.

## Appendix C. Implementation Details

### C.1. Network Architecture

- Input channels: 32 (8 wavelet + 24 condition wavelet)
- Output channels: 8 (wavelet subbands)
- Base channels: 64
- Channel multipliers: (1, 2, 2, 4, 4)
- Residual blocks per level: 2
- Normalization: GroupNorm with 32 groups
- Total parameters: 81,512,072

### C.2. Training Protocol

At each training step, we randomly select one of the four modalities as the target y, with the remaining three serving as conditions. This ensures balanced exposure to all synthesis directions. The model receives no explicit information about which specific modalities are conditioning inputs, only their wavelet representations and the target class label.

- Optimizer: AdamW ($\beta_1 = 0.9$, $\beta_2 = 0.999$)
- Learning rate: $10^{-5}$
- Batch size: 4
- Iterations: 50,000
- Gradient clipping: max norm 1.0
- Timestep sampling: $t \sim \mathcal{U}(0, 1)$
- Regularization noise: $\sigma = 0.5$

### C.3. Inference Protocol

For Heun's method with NFE=2:

- Timesteps: $t \in \{0, 0.5, 1.0\}$
- Step size: $\Delta t = 0.5$

**Timing (NVIDIA A100):**

- Single forward pass: 160ms
- Euler NFE=1: $160 \pm 0.05$ms (1 call)
- Heun NFE=1: $321 \pm 2.0$ms (2 calls)
- Heun NFE=2: $640 \pm 0.3$ms (4 calls)
- cWDM (1000 steps): 160s

## C.4. Per-Modality and Integration Analysis

**Per-modality analysis.** Performance varies substantially across target modalities. T1c synthesis shows the smallest gap (0.78 dB), likely because contrast enhancement patterns in T1c are partially predictable from the non-contrast T1 signal combined with T2/FLAIR tumor delineation. T1 synthesis shows the largest gap (2.12 dB), suggesting that reconstructing T1's specific intensity relationships from T1c/T2/FLAIR is more challenging than the reverse direction.

**Integration method comparison.** Heun's method with 2 steps (NFE=4) provides marginal improvement over single-step Euler (NFE=1): 26.80 vs. 26.72 dB average PSNR. This confirms that the learned velocity field is approximately constant; additional integration steps yield diminishing returns because the true trajectory is nearly linear.

## Appendix D. Additional Qualitative Results

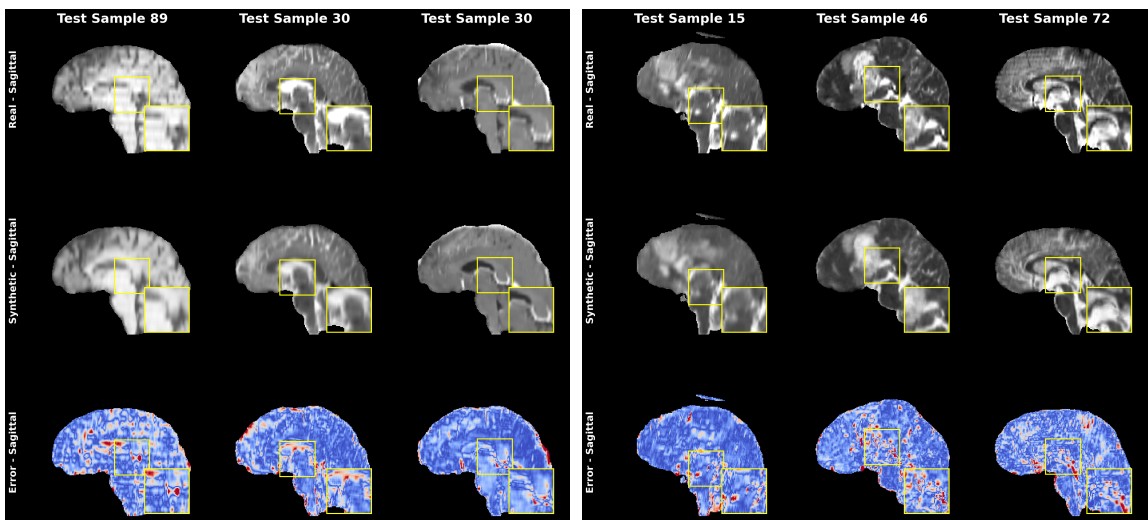

Figure 3: Qualitative results on sagittal slices with zoomed regions and error maps. Error maps use a blue-to-red colormap (red = higher error).

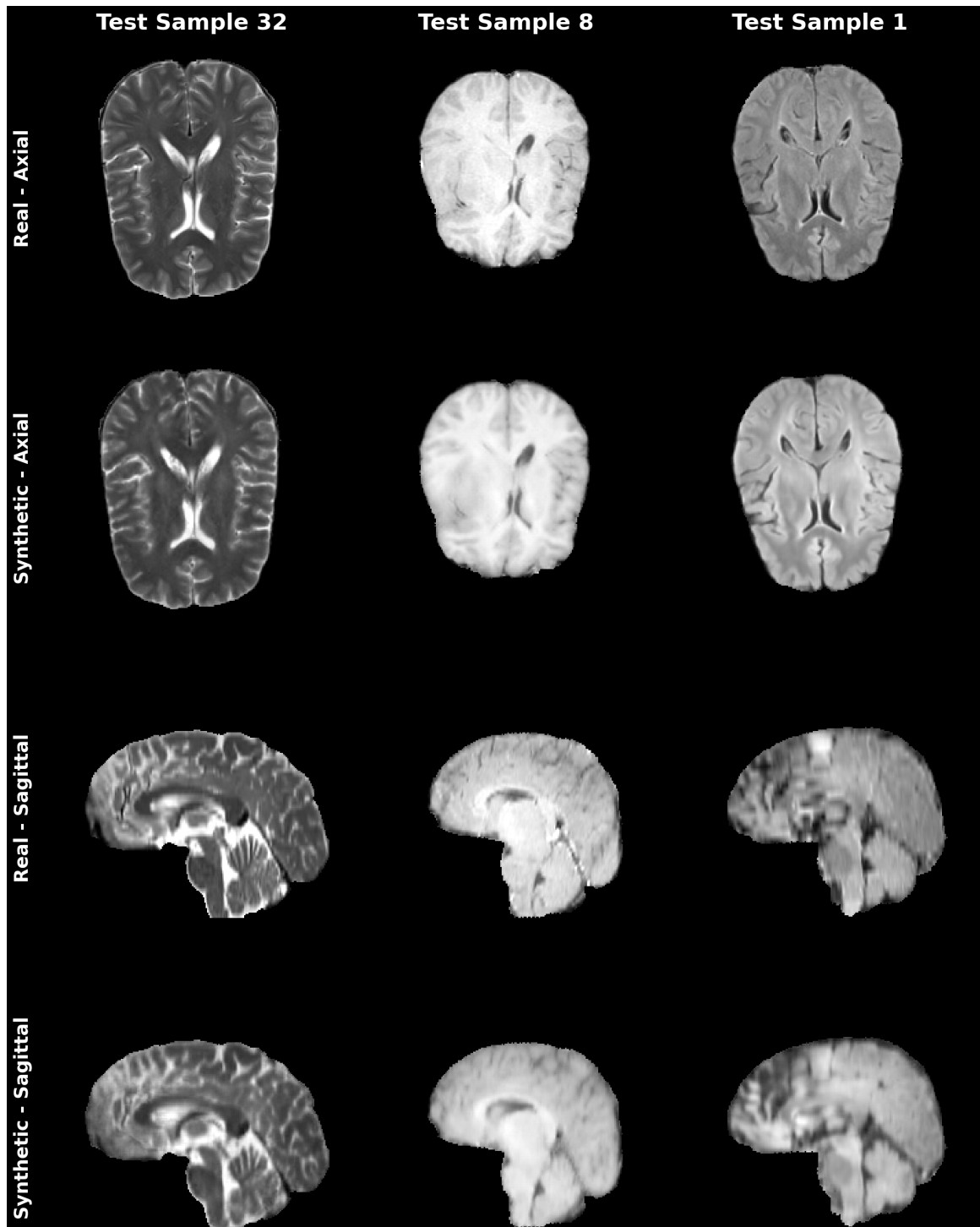

Figure 4: Additional qualitative results (samples 32, 8, 1). The method generalizes across the validation set while preserving structural details.

## Appendix E. Extended Discussion

**Regional error analysis.** Across the validation set, synthesis errors concentrate in three regions: 1. Tumor cores: As discussed above, enhancement unpredictability leads to under-detailed cores. 2. Skull boundaries: The sharp intensity transition at the skull-brain interface occasionally produces ringing artifacts in synthesized T1/T1c. 3. CSF spaces: FLAIR synthesis shows the highest error in CSF-containing regions (ventricles, sulci), where the CSF suppression effect is difficult to predict precisely from T1/T2 contrast.

**Flow Matching vs. Alternative Formulations** Several recent methods exploit informative priors for efficient generation. I$^2$SB (Liu et al., 2023) formulates image-to-image translation as a Schrödinger bridge between source and target distributions. BBDM (Li et al., 2023) uses Brownian bridge processes for the same purpose. LBM (Chadebec et al., 2025) operates in latent space with bridge matching. How does WFM relate to these approaches?

The key distinction is simplicity. Flow matching with a linear interpolation path yields a constant velocity field, enabling single-step inference without complex ODE solvers or bridge-specific training objectives. While I$^2$SB and BBDM achieve strong results, they typically require 10-50 steps for optimal quality. WFM trades theoretical generality for practical efficiency: by assuming that source and target are related by a simple contrast transformation, we obtain a formulation where one-step integration is sufficient.

This assumption is well-matched to multi-modal MRI synthesis but may not hold for more distant translation tasks (e.g., CT-to-MRI, where anatomical correspondence is weaker). For such settings, bridge-based methods with their greater flexibility may be preferable.

**Wavelet vs. Latent Space Processing** WFM operates in wavelet space rather than the latent space used by methods like Adaptive Latent Diffusion (Kim and Park, 2024). This choice has specific trade-offs.

Wavelet decomposition is invertible and deterministic: the 3D Haar transform compresses spatial dimensions by $2\times$ in each axis while preserving all information across eight subbands. There is no learned encoder, no reconstruction loss, and no risk of information bottleneck. The model operates directly on image content, not on a learned abstraction.

Latent diffusion, by contrast, uses a pretrained autoencoder to compress images into a lower-dimensional space. This can yield greater compression ($8\times$ or more) but introduces reconstruction error and requires the autoencoder to generalize across the target domain. For medical imaging, where subtle lesion details matter, wavelet space provides a more conservative choice that guarantees lossless reconstruction.

The 4x parameter reduction in WFM (82M vs. 326M for four cWDM models) comes not from aggressive compression but from weight sharing across modalities, a benefit of the unified architecture rather than the wavelet representation itself.

**The Quality-Speed Trade-off** WFM achieves 26.8 dB average PSNR compared to cWDM's 28.4 dB, a gap of approximately 1.6 dB. Is this trade-off acceptable?

The answer depends on the application. For real-time clinical review, where a radiologist needs to visualize a missing modality during a reading session, sub-second synthesis with 0.94 SSIM may be preferable to waiting 160 seconds for marginally higher fidelity. For

research applications requiring maximum accuracy (such as training segmentation models on synthetic data), the quality gap may matter more.

Notably, the gap is not uniform across modalities. T1 synthesis (27.6 dB) approaches cWDM quality (29.7 dB), while FLAIR shows a larger gap (26.1 vs. 27.8 dB). This modality dependence likely reflects the varying difficulty of contrast prediction: FLAIR's CSF suppression creates intensity patterns that are less predictable from T1/T2/T1c combinations.

Additional limitations include:

- **Single dataset evaluation.** All experiments use BraTS 2024, which consists of glioma patients. Generalization to other pathologies (metastases, stroke, multiple sclerosis) and healthy anatomy remains unvalidated.

- **No downstream task evaluation.** We report PSNR and SSIM but do not evaluate whether synthetic modalities improve tumor segmentation accuracy. This downstream validation is essential for clinical utility claims. In the future work, we will conduct segmentation as a downstream task.

- **Fixed conditioning configuration.** WFM assumes exactly three conditioning modalities are available. Handling variable numbers of inputs (one, two, or three available sequences) would require architectural modifications.

- **Comparison limited to diffusion.** We benchmark against cWDM but not against GAN-based methods (Pix2Pix, CycleGAN, ResViT) or other flow-based approaches. A broader comparison would strengthen the efficiency claims.

**Clinical Deployment Considerations** The 250-1000x speedup has practical implications beyond raw throughput. At 0.16-0.64 seconds per volume, WFM can run on-demand during clinical sessions rather than requiring batch processing. This enables interactive workflows: a radiologist reviewing an incomplete study could request synthesis of missing modalities in real time.

However, clinical deployment raises additional considerations not addressed in this work. Regulatory approval requires extensive validation across diverse patient populations. Uncertainty quantification (knowing when a synthesis is unreliable) is essential for safe clinical use. And integration with clinical PACS systems demands engineering effort beyond the algorithmic contribution.

We view WFM as a step toward clinically viable synthesis rather than a complete solution. The speed barrier has been addressed; the validation and integration challenges remain.

