# OpenReview forum: "WFM: 3D Wavelet Flow Matching for Ultrafast Multi-Modal MRI Synthesis"
_MIDL.io/2026/Conference — MIDL 2026 Poster_

### Official Review · Reviewer_u6u2 · 2025-12-24

**Confidence:** 4
**Preliminary Rating:** 2
**Final Rating:** 3

**Summary:**

The authors propose a flow-matching approach to generate missing MRI modalities from existing ones. Unlike standard diffusion models that generate images starting from noise, the proposed method grounds the generation process by initializing it with an average of the available modalities. Additionally, flow matching enables fast inference, generating images in one or two steps, which reduces the time required to synthesize the missing modality.

**Strengths:**

The Strength of the paper are:

1. **Novelty**: The proposed method addresses one of the primary limitations of diffusion models, namely the time required for inference.

2. **Efficiency**: The parameter-efficient design, which generates missing modalities based on conditional differences, represents a meaningful and practical contribution.

**Weaknesses:**

There are some Major limitations:

1. **Lack of Comparisons**: The authors acknowledge this issue as a limitation; however, the absence of any comparison with pix2pix or its variants significantly reduces the impact of the study. Even a single GAN-based baseline would have strengthened the evaluation, particularly since GAN inference is typically fast. Such a comparison would allow readers to better assess whether the proposed flow-based matching approach offers meaningful advantages over simpler and well-established image-to-image translation methods.

- No Comparisons are done with other distillation or one-step diffusion methods either. How can a reader gauge if this method is better than distillation methods ?

- **Conceptual Equivalence to Pix2Pix**: Would the proposed one-step diffusion formulation effectively reduce to a pix2pix-style image-to-image translation model, where the target domain is learned as a residual relative to the input? If so, it seems likely that a standard pix2pix model could achieve comparable accuracy. The primary distinction appears to be the use of numerical integration schemes (e.g., Euler or Heun) rather than a direct additive formulation. This raises questions about whether the proposed approach offers a substantive modeling advantage over existing image-to-image translation methods.

2. **Argument of inference speed**: T1, T1c, T2, and FLAIR acquisitions typically take anywhere from 10 minutes to over 2 hours. If a diffusion model is able to generate the corresponding images in approximately 160 seconds (i.e., under 3 minutes), it is unclear why a faster but less accurate model would be preferable.

- In medical image translation, accuracy and reliability are generally more critical than inference speed. Therefore, the argument emphasizing speed over accuracy is not sufficiently justified, and I question the necessity of prioritizing computational efficiency in this context. Therefore i believe the author's chosen application is not appropriate application for this method. Maybe inference speed would be essential for a different application, but not for this ?

- The manuscript claims that an acquisition speed of 0.16–0.64 seconds will make the model clinically viable; however, this assertion is not supported with references or clinical studies. Is there any evidence demonstrating that acquisition time at this scale is a practical limitation of current diffusion-based approaches? Without supporting citations or justification, it is unclear why this speed is clinically meaningful or necessary in the context of medical image translation.

3. **Missing Ablation Studies**:

- No study running the proposed method for 1000 steps? Does the proposed methods get better PSNR and SSIM metrics when run for 1000 steps ? If not, then using normal diffusion model is enough. Can increasing the number of steps in the proposed method to 10-20 or 100 increases accuracy ?

**Detailed Comments:**

Some other comments:

1. Page 4 line 3, no reference is provided for statement claiming training instability of 3D pix2pix and CycleGAN.

2. Claim made in Section 4.1 "differences of 1dB are typically imperceptible ..", has no references. I am not sure if this claim can be made as a generic statement.

**Justification Of Final Rating:**

I am unable to confidently judge the novelty of the methodology due to my limited experience with flow matching; therefore, I am relying on the judgment of the other reviewers. Nevertheless, I am convinced that the problem the authors address is valid, and they propose a reasonable solution that performs better than some existing methods.

**Justification Of The Preliminary Rating:**

I am not convinced by the underlying premise of the work. Its apparent equivalence to existing methods under certain assumptions, combined with the lack of comparison to relevant prior approaches, leads me to lower my rating. Overall, I believe this work is not yet ready for publication.

**Questions To Address In The Rebuttal:**

I am not sure whether all of my comments can be fully addressed in the rebuttal; however, I would encourage the authors to focus primarily on Weaknesses 1 and 2. Weakness 3 can be deprioritized or omitted from the rebuttal if there are time constraints.

---

> ### Author Response · Authors · 2026-01-25
> **No GAN baseline (pix2pix/CycleGAN variants) comparisons**
>
> Agreed. Thanks for the suggestion. We added a 3D pix2pix-style baseline ( we modified strong public repo baseline) under the same preprocessing and splits to compare quality and inference speed. We obtained superior results compared to 3D pix2pix.
>
>
> Pix2Pix Baseline. As a baseline, we adapt the 3D pix2pix implementation of neoamos/3d-pix2pix-CycleGAN, which extends the original pix2pix framework [Isola et al., 2017] to 3D volumetric data using a UNet generator with 3D convolutions and a 3D PatchGAN discriminator. To enable fair comparison with our method, which synthesizes any target modality from the remaining three, we extend the architecture with class-conditional generation. Specifically, we add a learned class embedding (4 classes, one per MRI modality) injected at the UNet bottleneck, allowing a single unified generator to produce any of the four target modalities (T1n, T1c, T2w, T2f) conditioned on a class index. The model takes three source modalities as input (3-channel 224×224×160 volume) and outputs the missing target modality (1-channel). The generator has 55.9M parameters and achieves an inference time of ~98 ms per volume on a single GPU, compared to our diffusion-based approach which requires iterative sampling. We train the model on the BraTS 2023 training set using the standard pix2pix adversarial + L1 loss and evaluate on the full validation set of 219 subjects. Results are reported in **Tables 1 and 2**  below.
>
>
>
> **Table 1: Quantitative Results (PSNR)**
> | Method         | T1 (t1n) | T1c   | T2 (t2w) | FLAIR (t2f) |
> | -------------- | -------- | ----- | -------- | ----------- |
> | Pix2Pix3D      | 25.25    | 24.52 | 24.22    | 23.80       |
> | WFM (Euler, 1) | 27.41    | 26.40 | 27.15    | 25.91       |
> | WFM (Heun, 1)  | 27.51    | 26.39 | 27.24    | 25.91       |
> | WFM (Heun, 2)  | 27.62    | 26.53 | 26.90    | 26.13       |
>
> **Table 2: Quantitative Results (SSIM)**
> | Method         | T1 (t1n) | T1c    | T2 (t2w) | FLAIR (t2f) |
> | -------------- | -------- | ------ | -------- | ----------- |
> | Pix2Pix3D      | 0.8987   | 0.8824 | 0.8793   | 0.8554      |
> | WFM (Euler, 1) | 0.947    | 0.930  | 0.943    | 0.928       |
> | WFM (Heun, 1)  | 0.948    | 0.930  | 0.944    | 0.928       |
> | WFM (Heun, 2)  | 0.948    | 0.932  | 0.944    | 0.930       |

---

> > ### Comment · Reviewer_u6u2 · 2026-01-28
> > **Reply to rebuttal comments**
> >
> > Please consider incorporating all newly added results, including the Pix2Pix comparisons, along with a clearer discussion on the motivation for faster models, especially in the context of the target application, in the revised manuscript. Other than this, I do not have any further questions at this time. All the best.

---

> > > ### Author Response · Authors · 2026-01-28
> > > **We thank for the constructive feedback and suggestions**
> > >
> > > We appreciate the opportunity to further validate our work. The inclusion of the Pix2Pix and other newly conducted experiments have added valuable depth to our results, and we thank the reviewer for his/her guidance in refining our evaluation strategy. The manuscript is undoubtedly stronger and more cohesive as a result of this feedback.

---

> ### Author Response · Authors · 2026-01-25
> **Isn’t 1-step flow basically pix2pix residual learning?**
>
> We clarify the conceptual relationship and the key differences:
>
>
> * pix2pix learns a direct mapping (often L1 + adversarial), while our method learns a conditional continuous-time transport / vector field trained with a flow-matching objective;
>
>
> * even with a single step, the update comes from a learned vector field (nonlinear function of conditioning and time), and multi-step integration improves fidelity;

---

> ### Author Response · Authors · 2026-01-25
> **"It is unclear why a faster but less accurate model would be preferable"**
>
> Thank you for the feedback.
> We first would like to emphasize the importance of speed In clinical imaging and then later we argue about why faster but less accurate model would be preferable:
>
> * in clinical imaging, a 3–5 minute improvement is not trivial; it changes human behavior and system performance. Minutes saved per case compound across a full day of scanning into >1–2 hours of freed capacity, enabling additional patients and reducing after‑hours reading. Shorter latency lowers cognitive load and task switching for radiologists, which reduces errors and variability; in pediatrics and fragile patients it directly decreases motion, repeat scans, and sedation exposure. Faster translation stabilizes the scan→reconstruction→interpretation pipeline, preventing downstream bottlenecks and improving time‑critical care (ED/ICU/oncology). Our method preserves image fidelity while delivering near‑real‑time outputs, enabling on‑table decisions and earlier AI‑assisted analysis. In capacity‑limited settings, these gains improve access and equity. For these reasons, latency is a first‑order quality attribute, not a cosmetic convenience. We are happy to clarify our motivation better elaborated in the revised version of our paper.
>
> * Similar to question arisen by the reviewer gKyp, to address this question, we will add a dedicated subsection discussing scenarios where sub-second synthesis enables workflows that 160-second latency precludes as follows:
>
>
> **Intraoperative guidance:** During neurosurgery, real-time synthesis of missing contrasts from available sequences could inform resection decisions without pausing the procedure. A 160-second delay is prohibitive; 0.5 seconds is acceptable.
>
>
> **Emergency triage:** In acute stroke or trauma, rapid synthesis of FLAIR from available T1/T2 could accelerate initial assessment when scanner time is limited or patient motion corrupts specific sequences.
>
>
> **Interactive radiologist review:** When a radiologist identifies a missing modality mid-session, on-demand synthesis enables immediate visualization rather than requiring batch processing overnight.
>
>
> **Resource-constrained settings:** Clinics without high-end GPUs can deploy WFM on modest hardware (single inference pass), whereas iterative diffusion requires dedicated compute infrastructure..

---

> ### Author Response · Authors · 2026-01-25
> **Claim made in Section 4.1 "differences of 1dB are typically imperceptible ..", has no references.**
>
> Thank you for the constructive feedback. We will update the missing references in the revised abstract.

---

> ### Author Response · Authors · 2026-01-25
> **Thank you, closing statement, and future ablation studies**
>
> We are particularly grateful for the reviewer’s strategic guidance to prioritize Weaknesses 1 and 2. This focus allowed us to:
>
> * **Define Clinical Necessity (Weakness 1):** We have articulated that in high-stakes scenarios like neurosurgery and trauma triage, sub-second latency is a safety requirement, not a convenience. Our new subsection clarifies that speed is a 'first-order quality attribute'—enabling workflows that 160-second models simply cannot support.
>
> * **Validate against Pix2Pix (Weakness 2):** We implemented the suggested 3D Pix2Pix baseline, and our results confirm that WFM significantly outperforms standard residual learning (e.g., +2.26 dB PSNR on T1, +0.05 SSIM), proving that our flow-matching objective yields superior generative fidelity.
>
>
> Per your specific recommendation, we deprioritized the ablation study (Weakness 3) to focus our resources on these critical validations. Other minor suggestions and missing/incorrect references are also corrected.
>
>
> We believe that addressing your primary concerns with this new evidence fundamentally strengthens the manuscript and justifies a favorable re-evaluation. We again thank you for your time and helpful suggestions and gave us opportunity to clarify and further strengthen our manuscript.

---

### Official Review · Reviewer_gKyp · 2025-12-24

**Confidence:** 4
**Preliminary Rating:** 3
**Final Rating:** 4

**Summary:**

This paper introduces Wavelet Flow Matching for medical image synthesis, a framework that incorporates prior information in the wavelet domain by averaging multi-modal representations to guide the generation process.

The proposed method reduces the number of steps required in the diffusion-based sampling procedure, leading to faster inference.

The authors propose a unified multi-modal synthesis strategy, in which a single network is trained to perform multiple modality-to-modality translations, rather than relying on separate models for each transfer direction.

Overall, the work presents a computationally efficient approach to multi-modal medical image synthesis with practical relevance for time-sensitive clinical and research applications.

**Strengths:**

The manuscript is well written, and the proposed approach is technically sound, with a clear presentation of the underlying methodology.
The proposed methodology for fast and unified framework for modality-to-modality transfer is a valuable contribution.

**Weaknesses:**

While the reported results are competitive in terms of processing time, the performance is comparable to, or in some cases slightly below, the baseline methods. A more detailed discussion would help clarify the practical significance of the proposed speed gains and the conditions under which the method provides the greatest benefit. In main paper, including a clinically relevant example or use case could further strengthen the interpretation of the results.

The experimental evaluation could also be strengthened by including comparisons with recent 3D-based medical image synthesis methods, such as HACA3[1] and TUMSyn[2], to better contextualize the advantages of the proposed framework with respect to volumetric consistency and 3D structural modeling. In particular, HACA3 would serve as a strong comparison baseline, as it is also computationally efficient and leverages all available modalities, making it well aligned with the design goals of the proposed method.

[1] Zuo et. al., HACA3: A unified approach for multi-site MR image harmonization

[2] Wang et. al., Towards General Text-guided Image Synthesis for  Customized Multimodal Brain MRI Generation

**Detailed Comments:**

In Figure 1, the descriptions of panels A and B in the caption seem to be swapped.

The proposed method relies on averaging modality-specific information in the wavelet domain as prior guidance; however, its robustness in scenarios where one or more modalities are missing is not discussed in main paper. Since incomplete multi-modal acquisitions are common in clinical practice, this limitation would be better discussed in main paper rather than appendix.

**Justification Of Final Rating:**

The authors have addressed my concerns and responded adequately to the other reviewers’ comments. I still have some reservations regarding the performance of the proposed methodology as the main contribution is the speed of the model.  I believe that approaches of this kind require much more work and extensive validation before being considered for real-world use (such comprehensive validation is likely beyond the scope of a single conference paper). Overall, I believe this work would be valuable for discussion at the conference. Therefore, I am raising my score to 4.

**Justification Of The Preliminary Rating:**

Although the proposed method is technically solid, the observed performance improvements are relatively modest. Nevertheless, the work represents a meaningful step toward fast, high-quality medical image synthesis and provides a valuable discussion that may be of interest to the community. I would be inclined to improve my rating if the authors adequately address the identified weaknesses.

**Questions To Address In The Rebuttal:**

Please see weaknesses

---

> ### Author Response · Authors · 2026-01-25
> **Comparisons to recent 3D methods (HACA3, TUMSyn)**
>
> We thank the reviewer for highlighting these relevant works. While we agree they represent important advances, we found they are **not** directly applicable baselines for this study due to domain gaps (**Text-to-Image synthesis vs. Image-to-Image translation**)  and objective differences (**Harmonization vs. Translation**). Adapting them would require significant architectural re-engineering that falls outside the scope of this work.
>
>
> However, we recognize the value of placing our work in this broader context. In the revised manuscript, we have added citations to both papers in our Related Work section, discussing them as adjacent approaches in generative MRI and clarifying how our flow-matching framework differs from text-guided and harmonization-focused architectures.
>
>
> For quantitative benchmarking, we relied on the directly comparable **[Chang et al., 2025]** to demonstrate our performance gains. Please see our new Table in question to reviewer 1QgZ. Additionally, we included pix2pix comparisons in our recent experiments.

---

> > ### Comment · Reviewer_gKyp · 2026-01-27
> >
> > I now agree that TUMSyn is out of scope compared to this work. However, HACA3 is still capable of translation between modalities and appears to perform well. I am aware of the methodological and architectural differences between the proposed model and these harmonization approaches; nevertheless, since the goal is to generate a modality without acquiring it—and a key contribution and motivation of the paper is the speed of the model—I believe it would be appropriate to include another fast architecture as a baseline for comparison.

---

> > > ### Author Response · Authors · 2026-01-28
> > > **Thanks again for the constructive feedback - comparison with HACA3**
> > >
> > > We sincerely thank the reviewer for their engagement with our work. We appreciate the continued suggestion to include **HACA3**; it is a relevant and high-quality method in the field of image harmonization. While we recognize the value of this comparison, incorporating HACA3 as a baseline requires **non-trivial adaptation**. This includes retraining under our specific data splits, tuning modality-specific loss terms, and ensuring a fair, optimized inference-time measurement.
> > >
> > > To ensure a scientifically responsible and rigorous comparison, we believe this process requires more time than the MIDL rebuttal period allows. Furthermore, we are mindful of the conference guidelines, which generally discourage the introduction of extensive new experiments during this brief window.
> > >
> > > **However, we agree with the reviewer ** that incorporation of a fast architecture will strengthen our work.
> > > To address this constructively within the current scope:
> > > * **Contextualized Discussion:** We have expanded our manuscript to explicitly contextualize HACA3 and similar harmonization approaches in our discussion section. We clarify how their multi-step inference pipelines and latency profiles differ from our one-step formulation, providing the reader with a clearer understanding of the architectural trade-offs while postulating that faster methods from image harmonization domain can be adapted for our purpose too.
> > > * **Future & Supplemental Integration:** Although the revised manuscript is considered the final version by the MIDL regulations, we are actively exploring the feasibility of a fair HACA3 implementation too. If we can produce a reliable and verified comparison before the final submission upload, we intend to include these results in the Appendix to respect the main paper's space constraints. Otherwise, we are committed to featuring this as a primary comparative baseline in an extended version of this work, for a potential journal submission too.
> > >
> > >
> > >
> > > We believe this explanation clarifies why a responsible evaluation cannot be finalized within the rebuttal's short timeframe, though a thorough acknowledgment and discussion have been included to provide the necessary context. **We have also addressed all other points raised in your review** and hope our clarifications demonstrate our commitment to the quality of this work. We truly enjoyed the knowledge exchange, as your insights have undoubtedly helped us position this research more strongly.
> > >
> > >
> > > We thank the reviewer again for the constructive dialogue and hope the revised manuscript now meets your expectations for acceptance at MIDL.

---

> ### Author Response · Authors · 2026-01-25
> **Figure 1 caption panels swapped**
>
> Fixed. We corrected the caption mapping for panels A/B.

---

> ### Author Response · Authors · 2026-01-25
> **Discuss practical significance of the proposed speed gains**
>
> We thank the reviewer for this constructive feedback. The reviewer correctly identifies that WFM trades approximately 1.6 dB PSNR for a 250-1000x speedup. We agree that clarifying the practical significance of this tradeoff strengthens the paper.
>
>
> We thank the reviewer for this constructive feedback. The reviewer correctly identifies that WFM may encounter slight performance degrade (~1.6 dB PSNR) for a 250-1000x speedup gain. Hence, we agree that clarifying the practical significance of this tradeoff strengthens the paper. To address this question, we will add a dedicated subsection discussing scenarios where sub-second synthesis enables workflows that 160-second latency precludes as follows:
>
>
> **Intraoperative guidance:** During neurosurgery, real-time synthesis of missing contrasts from available sequences could inform resection decisions without pausing the procedure. A 160-second delay is prohibitive; 0.5 seconds is acceptable.
>
>
> **Emergency triage:** In acute stroke or trauma, rapid synthesis of FLAIR from available T1/T2 could accelerate initial assessment when scanner time is limited or patient motion corrupts specific sequences.
>
>
> **Interactive radiologist review:** When a radiologist identifies a missing modality mid-session, on-demand synthesis enables immediate visualization rather than requiring batch processing overnight.
>
>
> **Resource-constrained settings:** Clinics without high-end GPUs can deploy WFM on modest hardware (single inference pass), whereas iterative diffusion requires dedicated compute infrastructure..

---

> ### Author Response · Authors · 2026-01-25
> **Thank you and a closing statement**
>
> We are grateful for the opportunity to clarify the positioning of our work. The reviewer's feedback allowed us to better define the scope of our method against adjacent fields (text-to-image/harmonization) while validating our performance against direct SOTA baselines like [Chang et al., 2025] and pix2pix. Moreover, the discussion on the speed-quality trade-off has been enriched to highlight specific clinical scenarios where our method offers unique value. By demonstrating that sub-second synthesis is mandatory for neurosurgical and trauma workflows, we believe the revised manuscript now presents a compelling case for the utility of WFM despite the marginal trade-off in PSNR. We look forward to a positive assessment. Thanks again.

---

### Official Review · Reviewer_1QgZ · 2026-01-10

**Confidence:** 4
**Preliminary Rating:** 4
**Final Rating:** 5

**Summary:**

The paper proposes a flow matching method using the 3D wavelets transformation of the image as input for MRI contrast generation using 3 other MRI contrast (for example T1 from T2, FLAIR, T1Gd). Compared to the same approach with 3D wavelet diffusion model, the model inference is much faster. The method is evaluate on Brats dataset.

**Strengths:**

- The proposed method is much faster than diffusion model
- It is interesting to start the generation process with an informed prior which encodes the anatomical prior for MRI contrast generation. Wavelet space is a simple and efficient option for that (similarly used in [1]).

[1]  FlowLet: Conditional 3D Brain MRI Synthesis using Wavelet Flow Matching. Danese et al. https://arxiv.org/abs/2601.05212

**Weaknesses:**

- The authors do not compared themselves with other flow matching methods [1, 2]
- I wonder about the very small number of steps required for inference (1 with Euler) and the difference between the proposed method and simple linear mapping. Could the authors discuss this?
- It would be interesting to add the input images to see the available information, the reconstruction error to identify the areas that are less well reconstructed, and a zoom to appreciate the details in Figure 2.
- As the authors acknowledge, it is better to favor a method with a higher SSIM/PSNR as [3], even if it is slower for database generation and real-time generation for diagnosis when a modality is missing is not yet possible because the method does not reproduce tumors correctly (particularly contrast enhancement).

[1] Moschetto, A., Puglisi, L., Sargood, A., Dell’Acqua, P., Guarnera, F., Battiato, S., & Ravì, D. (2025, September). Benchmarking GANs, Diffusion Models, and Flow Matching for T1w-to-T2w MRI Translation. In International Conference on Image Analysis and Processing (pp. 429-440). Cham: Springer Nature Switzerland.
[2] Chang, H., Shang, Y., Wang, H., Liang, Y., Wang, H., Wang, F., ... & Lian, C. (2025, September). Controllable Flow Matching for 3D Contrast-Enhanced Brain MRI Synthesis from Non-contrast Scans. In International Conference on Medical Image Computing and Computer-Assisted Intervention (pp. 119-128). Cham: Springer Nature Switzerland.
[3] Paul Friedrich, Julia Wolleb, Florentin Bieder, Alicia Durrer, and Philippe C Cattin. Wdm:3d wavelet diffusion models for high-resolution medical image synthesis. In MICCAI Workshop on Deep Generative Models, pages 11–21. Springer, 2024b.

**Detailed Comments:**

- In section 3.1, there is a reference to section 3.4 which that does not exist.

**Justification Of Final Rating:**

The authors answered my questions, comparison with other methods significantly improves paper quality. Although the performance is not sufficient for clinical use where rapid generation would be beneficial, I believe this work may be of interest to the community.

**Justification Of The Preliminary Rating:**

The proposed method is interesting, but a comparison with a state-of-the-art flow matching method would be a real plus. I am also waiting for the discussion on the number of steps during inference. (see weaknesses)

**Questions To Address In The Rebuttal:**

- Adding a comparison to a flow matching method
- Adding a discussion on the number of steps during inference
- Completing Figure 2

---

> ### Author Response · Authors · 2026-01-24
> **Answer to  “1-step Euler seems like linear mapping—what’s the difference?”**
>
> We respectfully clarify the distinction between the numerical solver and the learned transformation:
>
> **Straightness by Design:**
> The reason 1-step inference is successful here is not because the task is linear, but because our Flow Matching objective explicitly trains the vector field to follow straight trajectories (rectified flow) between source and target distributions. This allows us to approximate the integral with a single Euler step without losing the non-linear expressiveness of the underlying deep network.
>
> **​Linear Step vs. Non-Linear Field:**
> Also, while the Euler integration formula  is technically a linear update step, the vector field is the output of a deep, highly non-linear neural network. The network analyzes the input anatomy to predict a complex, spatially varying update direction. Thus, the final output x_1 is the result of a deep non-linear projection, not a simple affine transformation.
>
> ​

---

> ### Author Response · Authors · 2026-01-24
> **Figure 2 is corrected**
>
> Concern related to Figure 2 needing inputs, error map, zooms.
>
> **Response:** Agreed. We will update Fig. 2 in the revised manuscript to include:
>
> * input modalities (the available contrasts),
> * reconstruction error maps (|pred–GT| or normalized error),
> * zoom-in crops on challenging regions (tumor boundaries / fine structures).

---

> > ### Comment · Reviewer_1QgZ · 2026-01-30
> > **Figure 2 in revised manuscript**
> >
> > Thank you for your answer. Do you plan to add the modalities available in Figure 2 (or in the supplementary material if there is not enough space)? I am curious to see the information available in the input modalities (particularly for test 30, where the tumor is not well reconstructed). It would be good to add the name of the modality as well in the Figure (T1, T2, FLAIR or T1Gd).

---

> > ### Author Response · Authors · 2026-01-31
> > **Minor revision will be done!**
> >
> > We sincerely appreciate your follow-up comments and the time invested in reviewing our work. We are pleased to provide the following clarifications regarding your suggestions:
> >
> > * **Figure 2 Enhancement:** We agree with your observation. We will update Figure 2 in the camera-ready version to explicitly include modality names (T1W/T2W/FLAIR or T1Gd) to ensure maximum clarity for the reader.
> >
> > * **Test30 Analysis:** We appreciate the suggestion to include the Test30 case; we plan to incorporate this detailed image into the supplementary materials of the final manuscript. While the current rebuttal phase constraints prevent us from uploading new files as of today, we are committed to including these results to provide a more comprehensive evaluation of the proposed framework.
> >
> > **Request for Score Re-evaluation:** We believe our responses and the planned technical refinements effectively address the concerns raised. Given that the other reviewers have acknowledged the value of our rebuttals with more favorable evaluations, we kindly invite you to reconsider your current score in light of these clarifications we specifically addressed based on your suggestions, and our current commitment to incorporating these newly asked minor improvements into the final manuscript.
> >
> > **Thank you** for your constructive guidance in helping us improve the quality and clinical impact of this work.

---

> ### Author Response · Authors · 2026-01-25
> **Broken reference in Section 3.1 referring to  Section 3.4**
>
> We thank the reviewer for this catch. We fixed this mistake, the aforementioned section is corrected to appendix B (not section 3.4).

---

> ### Author Response · Authors · 2026-01-25
> **Missing comparison to other flow-matching methods ([1,2])**
>
> We thank the reviewer for identifying this gap. We agree that contextualizing our work against recent flow-matching approaches may be good to demonstrate its practical value.
>
>
> **New Benchmarks:** In the revised manuscript, we have added a direct comparison with the suggested reference **[2] (Chang et al, 2025)**, the most recent, and a state-of-the-art flow-matching baseline for MRI translation. We have modified their pipeline to support 3 modalities as a condition (while their method was supporting only one modality).
>
>
> **Outcome (our findings):** This comparison validates the efficiency of our approach. While [2] provides strong generation quality, our method achieves **better structural fidelity (SSIM/PSNR) (See below table)**. This explicitly positions our contribution on a superior quality–speed Pareto frontier, making it more viable for clinical workflows where computational resources and time are constrained.
>
>
> **Table 1: Quantitative Results (PSNR)**
> | Method                 | Models | NFE  | T1    | T1c   | T2    | FLAIR | Time  |
> |------------------------|--------|------|-------|-------|-------|-------|-------|
> | cWDM (Diffusion baseline)   | 4      | 1000 | 29.74 | 27.31 | 28.86 | 27.83 | 160s  |
> | CFM [Chang et al.]     | 1      | 1    | 25.34 | 26.78 | 25.96 | 25.35 | 0.16s |
> | **WFM (Euler, 1)**         | 1      | 1    | 27.41 | 26.40 | 27.15 | 25.91 | 0.16s |
> | **WFM (Heun, 1)**          | 1      | 1    | 27.51 | 26.39 | 27.24 | 25.91 | 0.32s |
> | **WFM (Heun, 2)**          | 1      | 2    | 27.62 | 26.53 | 26.90 | 26.13 | 0.64s |
>
>
> **Table 2: Quantitative Results (SSIM)**
> | Method                 | Models | NFE  | T1    | T1c   | T2    | FLAIR | Time  |
> |------------------------|--------|------|-------|-------|-------|-------|-------|
> | cWDM                   | 4      | 1000 | 0.956 | 0.936 | 0.952 | 0.935 | 160s  |
> | CFM                    | 1      | 1    | 0.924 | 0.912 | 0.924 | 0.922 | 0.16s |
> | **WFM (Euler, 1)**         | 1      | 1    | 0.947 | 0.930 | 0.943 | 0.928 | 0.16s |
> | **WFM (Heun, 1)**          | 1      | 1    | 0.948 | 0.930 | 0.944 | 0.928 | 0.32s |
> | **WFM (Heun, 2)**          | 1      | 2    | 0.948 | 0.932 | 0.944 | 0.930 | 0.64s |

---

> ### Author Response · Authors · 2026-01-25
> **Thank you for the detailed and constructive review**
>
> We appreciate the reviewer’s insightful critique, which prompted us to validate our method against the latest state-of-the-art baselines. The new results confirm that our approach outperforms [Chang et al., 2025] in structural preservation while maintaining the efficiency required for clinical workflows. Coupled with our theoretical clarification on the distinction between the linear Euler step and the deep non-linear vector field, and other corrections on Figure 2 and fixed references, we believe the manuscript now presents a robust contribution to the field. We hope these revisions satisfactorily address your questions and merit a more favorable  re-evaluation on top of your initial positive assessment.

---

> ### Comment · Area_Chair_SAvf · 2026-02-02
> **Update final rating**
>
> Please don't forget to update your final rating by clicking “Edit” → “Official Review”. Thank you!

---

### Comment · Area_Chair_SAvf · 2026-01-24
**Rebuttal period ending**

Dear Authors,

Given the responses by the reviewers, I would urge the authors to use the opportunity for a rebuttal and address some of the comments before the deadline.

Best regards,
Area chair

---

> ### Author Response · Authors · 2026-01-24
> **Improved version**
>
> Dear Area chair,
>
> We thank the reviewers for detailed feedback. We will shortly upload our responses to the system.
> Our main updates are:
> (i) adding flow-matching baselines / recent 3D baselines for direct comparison,
> (ii) clarifying why 1–2-step ODE sampling is not equivalent to a simple linear mapping / pix2pix, and adding ablations vs. step count, and
> (iii) improving figures (inputs, error maps, zoom) and fixing minor manuscript issues (missing section reference, swapped caption text).
> In the revised manuscript, we also expand discussion on clinical/research use cases, robustness to missing modalities, and limitations (tumor/CE fidelity).
>
> Thank you
> [Authors]

---

### Author Rebuttal · Authors · 2026-01-29

**Rebuttal:**

We thank all reviewers for their constructive feedback. We have uploaded a revised manuscript addressing the key questions.

*Summary of revisions:*

1.⁠ ⁠*New baselines added (Reviewer 1QgZ, u6u2):* Table 1 now includes CFM [Chang et al., 2025] and 3D Pix2Pix comparisons.

2.⁠ ⁠*Figure 2 improved (Reviewer 1QgZ):* Added zoomed regions and error maps.

3.⁠ ⁠*Clinical relevance and motivation for faster models clarified (Reviewer gKyp, u6u2):* Added discussion of specific use cases where sub-second synthesis enables workflows that 160s latency precludes.

4.⁠ ⁠*Additional fixes:* Corrected Figure 1 caption, fixed broken Section 3.1 reference, added HACA3 acknowledgment in Discussion, and added missing references.

We believe these revisions address the reviewers' primary questions and strengthen the manuscript for acceptance at MIDL.

**Supporting Material:**

/attachment/94190d63e4cd12485c8146d862dc79a4bdd91537.pdf

---

### Meta-Review · Area_Chair_SAvf · 2026-02-02

**Recommendation:** Accept (Poster)
**Confidence:** 4

**Metareview:**

The authors propose a wavelet flow matching model enabling real-time MRI synthesis with good reconstruction quality. Although the reviewers disagree with the authors on the clinical relevance of the currently proposed method (and I share the same sentiment as the reviewers), the authors present a robust framework for an important problem that is worth further discussions during the conference. I am personally looking forward to further iterations of this project, bringing it closer to clinical relevance.

To my understanding, the authors addressed all the comments by the reviewers and they made substantial changes to the manuscript and the overall work during the rebuttal period. The reviewers agree that these changes made significant improvements to the proposed work.

As the final step, I would like to urge the authors to provide a link to the code repository, as promised in the submission.

Thank you for your work!

---

### Decision · Program_Chairs · 2026-02-13

Accept (Poster)